# Measurement-Based Quantum Thermal Machines with Feedback Control

**DOI:** 10.3390/e25020204

**Published:** 2023-01-20

**Authors:** Bibek Bhandari, Robert Czupryniak, Paolo Andrea Erdman, Andrew N. Jordan

**Affiliations:** 1Institute for Quantum Studies, Chapman University, Orange, CA 92866, USA; 2Department of Physics and Astronomy, University of Rochester, Rochester, NY 14627, USA; 3Center for Coherence and Quantum Optics, University of Rochester, Rochester, NY 14627, USA; 4Department of Mathematics and Computer Science, Freie Universität Berlin, Arnimallee 6, 14195 Berlin, Germany

**Keywords:** discrete quantum measurement, continuous quantum measurement, quantum feedback, Maxwell’s demon, refrigerator

## Abstract

We investigated coupled-qubit-based thermal machines powered by quantum measurements and feedback. We considered two different versions of the machine: (1) a quantum Maxwell’s demon, where the coupled-qubit system is connected to a detachable single shared bath, and (2) a measurement-assisted refrigerator, where the coupled-qubit system is in contact with a hot and cold bath. In the quantum Maxwell’s demon case, we discuss both discrete and continuous measurements. We found that the power output from a single qubit-based device can be improved by coupling it to the second qubit. We further found that the simultaneous measurement of both qubits can produce higher net heat extraction compared to two setups operated in parallel where only single-qubit measurements are performed. In the refrigerator case, we used continuous measurement and unitary operations to power the coupled-qubit-based refrigerator. We found that the cooling power of a refrigerator operated with swap operations can be enhanced by performing suitable measurements.

## 1. Introduction

The quest to invent a thermal machine at the nanoscale has led to the new field of quantum thermodynamics [1,2,3,4,5,6]. Thanks to recent advances in nanofabrication techniques, much attention has been focused on realizing nanoscale-based quantum devices [7,8,9,10,11,12,13,14,15] for heat management. Consequently, understanding how to control heat transport and dissipation at the nanoscale is of utmost significance and could enhance the performance of quantum devices’ power and efficiency. Within the field of quantum thermodynamics, quantum thermal machines, such as heat engines and refrigerators, have been theoretically and experimentally investigated in detail [2,16,17,18,19,20,21,22,23,24,25,26,27,28,29,30,31,32,33,34,35,36,37,38,39]. Quantum refrigerators are quantum devices where heat is extracted from a cold thermal bath. Usually, they are powered by external work provided by a chemical potential imbalance [3,40] or by external driving [41,42,43,44,45,46].

Quantum-limited measurements are now being performed regularly within the field of quantum computation. In contrast to classical measurements, quantum measurements can be “invasive”, i.e., they can change the system’s state and, consequently, the energetics of the system [47,48,49,50]. This leads to a change in the quantum device’s functioning and performance depending on the measurement type and strength [47,48,49,50,51,52,53,54,55]. In particular, in the case of quantum devices, it can be important to keep track of the quantum measurement outcomes and act on the system accordingly to achieve a given task.

Technological advancement has enabled the experimental realization of quantum thermal machines powered by measurements and feedback, such as Maxwell’s demons [8,56,57] and Szilard’s engines [7]. These are devices where measurements and feedback allow, respectively, the extraction of heat or work from a single thermal bath—apparently violating the second law of thermodynamics. These realizations have motivated further research in the field, leading to an entire family of quantum measurement and feedback-based thermal machines. Heat and work extraction has been studied in various quantum systems exploiting quantum measurements with different strengths (weak or projective) and natures (invasive or non-invasive) [49,50,58,59,60,61,62,63,64,65]. Although both invasive and non-invasive quantum measurements can be used to obtain information about the quantum system and run a feedback loop to power quantum thermal machines, it has been observed that invasive measurements alone can be used as the fuel to power a thermal machine [49,50,64].

A Maxwell’s demon powered by projective quantum measurements was studied in single-qubit systems in [57,66,67,68,69] and in double-quantum dot systems in [70]. Recently, it was observed that also weak quantum measurements can be employed to realize a single-qubit-based Maxwell’s demon and a refrigerator powered by invasive measurements and feedback [63]. Furthermore, quantum measurements have also been utilized to realize heat engines [71,72,73,74,75,76,77,78,79,80,81,82], qubit elevators [83], and quantum batteries [84,85], among other devices.

In this paper, we studied various configurations of coupled-qubit-based thermal devices, namely a *quantum Maxwell’s demon*, and a *measurement-assisted refrigerator*, the latter being a system that extracts heat from a cold bath exploiting the combination of external work and invasive quantum measurements. As opposed to previous literature, we considered coupled-qubit-based devices powered by weak quantum measurements, both discrete and continuous. We studied the performance of the machine in various configurations using different feedback strategies based on local measurements. In the Maxwell’s demon case, we compared the impact of performing simultaneous measurements of both qubits on a single setup and performing only individual qubit measurements on two setups operated in parallel. Thanks to a beneficial collective effect, we found that the former can outperform the latter. In the continuous measurement case, we computed the work distribution related to the stochasticity of the measurement outcome, allowing us to observe quantities, such as power fluctuations, that are beyond the average thermodynamic quantities. At last, in the refrigerator case, we show how the addition of invasive quantum measurements, in the absence of feedback, can enhance the performance of a refrigerator powered by external work. The results obtained in this paper for the case of a measurement-assisted refrigerator can be straightforwardly extended to the case of coupled-quantum dots attached to fermionic baths. In addition, the formulation used in this paper can be used to study finite-time statistics of different thermodynamic variables in terms of the measurement record, which can be directly accessible in an experiment [63].

The paper is organized as follows. In the next section, we introduce the models studied in this paper and the corresponding formalism. In Section 3, we study the coupled-qubit device operated as a Maxwell’s demon. We study both discrete and continuous measurements, as well as the impact of measuring a single qubit or both. In Section 4, we study the device operated as a measurement-assisted refrigerator under continuous measurements. In Section 5, we draw the conclusions.

## 2. Model

We considered the setup in Figure 1: two coupled-qubits, $Q_{1}$ and $Q_{2}$, are, respectively, coupled to two thermal baths at temperatures $T_{1}$ and $T_{2}$ and to two measurement apparatuses $D_{1}$ and $D_{2}$, which allows us to perform local quantum measurements on the respective qubits. The total Hamiltonian for the setup is given by $H=H_{Q}+H_{B}+H_{C}$, where the Hamiltonian of the coupled-qubit system is
(1)$$H_{Q}=\frac{\epsilon_{1}}{2}\sigma_{z}^{\left(1\right)}+\frac{\epsilon_{2}}{2}\sigma_{z}^{\left(2\right)}+\Delta_{x}\sigma_{x}^{\left(1\right)}\sigma_{x}^{\left(2\right)}+\Delta_{y}\sigma_{y}^{\left(1\right)}\sigma_{y}^{\left(2\right)}+\Delta_{z}\sigma_{z}^{\left(1\right)}\sigma_{z}^{\left(2\right)},$$$\epsilon_{i}$ being the qubit gap for qubit $Q_{i}$ and $\Delta_{i}$ the strength of the $\sigma_{i}$-$\sigma_{i}$ coupling between the two qubits. $H_{B}$ is the Hamiltonian describing the heat baths, which we considered as bosonic baths with continuous degrees of freedom:(2)$$H_{B}=\sum_{i=1,2}\sum_{k}\epsilon_{ik}b_{ik}^{†}b_{ik},$$
where $b_{ik}\left(b_{ik}^{†}\right)$ are the bosonic annihilation (creation) operators with energy $\epsilon_{ik}$ and quantum number *k* for bath *i*. We considered a linear “tunnel-like” coupling between the baths and the system given by
(3)$$H_{C}=\sum_{i=1,2}\sum_{k}V_{ik}\left(\sigma_{+}^{\left(i\right)}b_{ik}+b_{ik}^{†}\sigma_{-}^{\left(i\right)}\right),$$
where $\sigma_{\pm }^{\left(i\right)}$ are ladder operators for qubit $Q_{i}$.

We considered both discrete and continuous, as well as strong and weak quantum measurements. All such scenarios can be described by positive-operator-valued measures (POVMs), i.e., by a set of Krauss operators $M_{k}$, one for each measurement outcome, satisfying $\sum_{k}M_{k}^{†}M_{k}=I$ for the discrete case and $\int dkM_{k}^{†}M_{k}=I$ for the continuous case [63,86]. The specific form of the Krauss operators for discrete and continuous measurements will be discussed in Section 3. The probability (probability density in the continuous case) of measuring outcome *k* is given by $\mathrm{Tr}[\rho M_{k}^{†}M_{k}]$, where ρ is the reduced density matrix of the coupled-qubit system. The post-measurement state $\rho_{M_{k}}$, conditioned by observation *k* and assumed to occur instantaneously, is given by
(4)$$\rho_{M_{k}}=\frac{M_{k}\rho M_{k}^{†}}{\mathrm{Tr}[\rho M_{k}^{†}M_{k}]}.$$

Throughout this paper, we considered two operational regimes: the *quantum Maxwell’s demon* and the *measurement-assisted refrigerator*. In the quantum Maxwell’s demon case, we considered a single temperature of the environment, i.e., $T_{1}=T_{2}=T$. In this configuration, the aim is to extract heat from the single-temperature bath exploiting invasive quantum measurements and feedback. In the measurement-assisted refrigerator, we considered an environment consisting of two different temperatures $T_{1}$ and $T_{2}$, and the aim is to maximize the heat extracted from the cold bath. Here, the refrigerator is powered by a combination of work, delivered by an external control, and invasive quantum measurements in the absence of feedback.

## 3. Quantum Maxwell’s Demon

In this section, we describe our results operating the coupled-qubit-based thermal machine as a quantum Maxwell’s demon. Here, we only considered the $\sigma_{z}$-$\sigma_{z}$ coupling between the two qubits, i.e., $\Delta_{x}=\Delta_{y}=0$. In order to restrict the space of all possible quantum measurements and feedback strategies, we focused on local quantum measurements (i.e., using local probes $D_{1}$ and $D_{2}$, schematically shown in Figure 1) and local feedback strategies that are simple to implement experimentally.

In particular, we considered unitary feedback consisting of local single-qubit unitary rotations around the *y*-axis, i.e., of the form $U_{i}\left(\theta_{i}\right)=e^{-i\theta_{i}\sigma_{y}^{\left(i\right)}}$, where $\theta_{i}$ is a suitable angle. We considered both discrete and continuous quantum measurements of the spin state of each qubit in the x-direction. Discrete weak $\sigma_{x}$ measurements performed on qubit $Q_{i}$ using probe $D_{i}$, for i=1,2, are described by the operators $\left\{M_{i+},M_{i-}\right\}$, where
(5)$$M_{i\pm }=\frac{1}{2}\left(\sqrt{\kappa_{i}}+\sqrt{1-\kappa_{i}}\right)I_{2}⊗I_{2}\pm \frac{1}{2}\left(\sqrt{\kappa_{i}}-\sqrt{1-\kappa_{i}}\right)\cdot \left\{\begin{array}{c} \sigma_{x}^{\left(1\right)}⊗I_{2}\;\;\mathrm{for}\;i=1,\\ I_{2}⊗\sigma_{x}^{\left(2\right)}\;\;\mathrm{for}\;i=2, \end{array}\right.$$$I_{2}$ is the 2 × 2 identity, and $\kappa_{i}=1/2-\sqrt{2\gamma_{i}^{\prime }\delta t}$ is an indicator of the strength of the discrete measurement with characteristic measurement rate $\gamma_{i}^{\prime }$ and measurement time δt; these can be related to the resolution of the detector [47,87]. The k→0,1 limits describe strong (projective) measurements, where the demon acquires maximum information about the system, whereas k→1/2 describes the opposite limit, where no information is acquired. Intermediate values of *k* describe the transition from strong to weak measurements.

In the case of continuous measurement, we have a continuum of Krauss operators ${\left\{M_{i,r_{i}}\right\}}_{r_{i}}$, one for each measurement apparatus *i*, where $r_{i}$ is the continuous measurement outcome. They are given by
(6)$$\begin{array}{cc} \; & M_{1,r_{1}}={\left(\frac{\delta t}{2\pi \tau }\right)}^{\frac{1}{4}}\exp\{-\frac{\delta t{\left(r_{1}I_{2}-{\hat{\sigma }}_{x}^{\left(1\right)}\right)}^{2}⊗I_{2}}{4\tau }\},\\ \; & M_{2,r_{2}}={\left(\frac{\delta t}{2\pi \tau }\right)}^{\frac{1}{4}}\exp\{-\frac{\delta t\;I_{2}⊗{\left(r_{2}I_{2}-{\hat{\sigma }}_{x}^{\left(2\right)}\right)}^{2}}{4\tau }\}, \end{array}$$
where δt is the time allocated to perform a single measurement and τ is the characteristic measurement time scale taken to separate the two-measurement distribution by two standard deviations [53,63]. In other words, τ can be understood as the inverse of the measurement strength and is the time required to achieve the unit signal-to-noise ratio [53]. When δt/τ is large, the measurement is often referred to as strong measurement, whereas the measurements for which δt/τ is small are called weak measurements. Following Equation (Equation 6), the measurement readout is randomly sampled from two Gaussian distributions with variance $\sqrt{\tau /\delta t}$ and mean +1 (associated with the $\sigma_{x}=+1$ measurement outcome) and −1 (associated with the $\sigma_{x}=-1$ measurement outcome).

We operated the system as a Maxwell’s demon considering the following thermodynamic cycle, consisting of three strokes: (i) measurement, (ii) feedback, and (iii) thermalization:

(i) Assuming the system to be initialized in a thermal state $\rho_{T}=e^{-H_{Q}/\left(k_{B}T\right)}/Z$, where $Z=\mathrm{Tr}[e^{-H_{Q}/\left(k_{B}T\right)}]$, the initial energy of the coupled-qubits is given by
(7)$$E_{T}=\mathrm{Tr}\left[\rho_{T}H_{Q}\right].$$Let $\rho_{i}={\mathrm{Tr}}_{\tilde{i}}\left[\rho \right]$ be the single-qubit density matrices given by tracing out the other qubit, where $\tilde{i}=2$ ($\tilde{i}=1$) for i=1 (i=2). Notice that, in the thermal state $\rho_{T}$, the Bloch vectors of each single-qubit density matrix only have a *z* component, since $\Delta_{x}=\Delta_{y}=0$. A quantum measurement is now performed using either $D_{1}$ or both $D_{1}$ and $D_{2}$. After performing a measurement, the state changes to $\rho_{M_{k}}$. Now, the Bloch vector of the measured qubits acquires an *x* component, and the norm of the vector may change.

(ii) Feedback is performed by applying unitary rotations $U_{i}\left(\theta_{i}\right)$ around the y-axis to the qubits that have been measured. The angle $\theta_{i}$ is conditioned on the measurement outcome. Indeed, it was chosen such that the single-qubit states $\rho_{i}$, corresponding to the measured qubits, are rotated to the positive or negative z-axis of the Bloch sphere. The feedback that brings the state of $Q_{i}$ back to the positive (negative) z-axis will be denoted as $F_{i}=1$ ($F_{i}=-1$). The state after the measurement and feedback is given by $\rho_{F_{k}}=U_{1}\left(\theta_{1}\right)\rho_{M_{k}}U_{1}^{†}\left(\theta_{1}\right)$ if only $D_{1}$ is used and by $\rho_{F_{k}}=U_{2}\left(\theta_{2}\right)U_{1}\left(\theta_{1}\right)\rho_{M_{k}}U_{1}^{†}\left(\theta_{1}\right)U_{2}^{†}\left(\theta_{2}\right)$ if both detectors are used. The energy of the system after measurement and feedback is given by $E_{F_{k}}=\mathrm{Tr}\left[\rho_{F_{k}}H_{Q}\right]$. Notice that $E_{F_{+}}=E_{F_{-}}=E_{F}$.

(iii) The cycle is closed, allowing a full thermalization of the system with the thermal baths. During this stroke, the state of the system returns to $\rho_{T}$, and an amount of heat $Q=E_{T}-E_{F}$ is extracted from the bath.

### 3.1. Discrete One-Qubit Measurement

In this subsection, we only perform measurements with $D_{1}$. As a consequence, only the state $\rho_{1}$ of $Q_{1}$ changes. Let us denote with $x_{1}$ and $z_{1}$ the x and z components of the Bloch vector corresponding to $\rho_{1}$ after the measurement. The angle of the unitary rotation $U_{1}\left(\theta_{1}\right)$ corresponding to feedback $F_{1}=1$ is given by $\theta_{1}=-\frac{1}{2}\tan^{-1}\left(\frac{x_{1}}{z_{1}}\right)$, whereas for feedback $F_{1}=-1$, it is given by $\theta_{1}=-\frac{1}{2}\tan^{-1}\left(\frac{x_{1}}{z_{1}}\right)+\pi /2$. The angle is chosen to rotate the qubit to the positive ($F_{1}=1$) or negative ($F_{1}=-1$) *z*-axis.

In Figure 2, we investigate the heat *Q*, extracted from the heat bath, as a function of the qubit–qubit coupling strength $\Delta_{z}$ for different values of $\kappa_{1}$ and feedback strategies (Panel (a)) and as a function of the measurement strength $\kappa_{1}$ for different values of $\Delta_{z}$ (Panel (b) corresponding to feedback $F_{1}=1$ and Panel (c) to $F_{1}=-1$). In the case of decoupled-qubits, i.e., $\Delta_{z}=0$, we know that heat extraction can be obtained only with $F_{1}=-1$ [63], since $F_{1}=1$ would increase the energy of the qubit, resulting in heating the baths, rather than cooling them. However, for finite $\Delta_{z}$, we observe that positive heat extraction can be obtained even with $F_{1}=+1$ (see the solid black and dotted red curves in Figure 2a). The heat extraction, in this case, is obtained when $\Delta_{z}>0.1\;k_{B}T$. The behavior above can be explained by considering that the energetics of the coupled systems is influenced by $\Delta_{z}$. Note that, for a suitable choice of feedback, the heat extraction is an increasing function of $\Delta_{z}$ in the considered parameter regime. In Figure 2b,c, we observed that, for $\kappa_{1}=0.5$, i.e., when the demon acquires no information from the measurement, the qubit dissipates heat to the bath for all values of $\Delta_{z}$. Since no information is obtained, the demon has no resources to extract heat from a single thermal bath. Conversely, the maximum heat is extracted from the bath when $\kappa_{1}\to 0,1$, which corresponds to maximum information extraction (the demon performs a projective measurement and feedback). Comparing Figure 2b,c, we observed that changing the feedback strategy $F_{1}$ from +1 (Panel (b)) to −1 (Panel (c)) or vice versa changes the sign of heat extraction.

### 3.2. Discrete Two-Qubit Combined Measurement

In this section, we measure the state of the system using both $D_{1}$ and $D_{2}$ simultaneously. As a consequence, both $\rho_{1}$ and $\rho_{2}$ are affected by the measurement, so we will apply both $U_{1}\left(\theta_{1}\right)$ and $U_{2}\left(\theta_{2}\right)$ as feedback. Let us denote with $x_{i}$ and $z_{i}$ the x and z component of the Bloch vector corresponding to $\rho_{i}$ after measurement. The angle of the unitary operation that corresponds to feedback $F_{i}=1$ applied to $Q_{i}$ is given by $\theta_{i}=-\frac{1}{2}\tan^{-1}\left(\frac{x_{i}}{z_{i}}\right)$, whereas for feedback $F_{i}=-1$ applied to $Q_{i}$, it is given by $\theta_{i}=-\frac{1}{2}\tan^{-1}\left(\frac{x_{i}}{z_{i}}\right)+\pi /2$.

In Figure 3, we study the heat extraction (Q) out of the baths as a function of the qubit–qubit coupling strength $\left(\Delta_{z}\right)$. Since $\epsilon_{1}\ll \epsilon_{2},k_{B}T$, we observed that heat extraction is possible only for the feedback $F_{2}=-1$ on the qubit $Q_{2}$ (see the negative values of *Q* in the inset). However, the choice of feedback on the qubit $Q_{1}$ depends on the value of $\Delta_{z}$ (see the red and black curves). As opposed to the single-qubit case [63] (see Figure 2a), here, there are value of $\Delta_{z}$ where both feedback strategies $F_{1}=+1$ and $F_{1}=-1$ result in cooling. In the other limit, when $\epsilon_{2}\ll \epsilon_{1},k_{B}T$ (not shown in the figure), heat extraction can be obtained only with feedback $F_{1}=-1$ on $Q_{1}$. For $\epsilon_{1},\epsilon_{2}\gg k_{B}T$, we observed heat extraction only for the feedback $F=\left(F_{1},F_{2}\right)=\left(-1,-1\right)$, since the system effectively behaves as two decoupled-qubits.

We now study whether the combined use of both detectors $D_{1}$ and $D_{2}$ on a single-coupled-qubit system (“combined case”) can lead to a better performance with respect to having two-coupled-qubit systems operated in parallel where only $D_{1}$ is applied to one system and $D_{2}$ to the other one (“individual case”). Notice that, in this comparison, the number of measurements is the same. In Figure 4, we plot the extracted heat as a function of $\Delta_{z}$, comparing these two scenarios. The solid red curve corresponds to the individual case, whereas the dashed black curve corresponds to the combined case. Notably, for the set of parameters considered, we observed that the combined case can outperform the individual case. Interestingly, we noticed that the advantage of the combined case, i.e., the difference between the two curves, is enabled by the interaction between the qubits, and for $\Delta_{z}>0$, it increases monotonically with increasing interaction strength. For large values of $\Delta_{z}$, the state of the coupled-qubit system after feedback has larger energy compared to its initial thermal energy, leading to a heating effect instead of cooling.

### 3.3. Continuous One- and Two-Qubit Measurement

Continuous feedback, which is widely used in optimal control in classical systems, depends on the continuous input of the measurement record. Continuous-quantum-measurement-based feedback is a natural extension of the classical optimal control theory. In the quantum feedback theory based on continuous measurement, one studies the evolution of the density matrix under the influence of measurement and other external probes and suitably tunes the feedback control based on the continuous stream of the measurement record. The evolution of the density matrix based on the stream of the measurement record is referred to as a quantum trajectory [88,89]. Experiments utilizing continuous-measurement-based feedback have been realized on several platforms, including quantum optics [90] and quantum error correction [91].

In this subsection, we study the distribution of the extracted heat performing a cooling cycle as in the previous subsections, but replacing the discrete measurement with a continuous measurement. Using Equation (Equation 4), the state of the coupled-qubit system after measurement can be written as
(8)$$\rho_{M,r_{1}}=\frac{M_{1,r_{1}}\rho M_{1,r_{1}}^{†}}{\mathrm{Tr}[\rho M_{1,r_{1}}^{†}M_{1,r_{1}}]},$$
when measurement is performed only with $D_{1}$ and
(9)$$\rho_{M,r_{1}r_{2}}=\frac{M_{2,r_{2}}M_{1,r_{1}}\rho M_{1,r_{1}}^{†}M_{2,r_{2}}^{†}}{\mathrm{Tr}[\rho M_{2,r_{2}}^{†}M_{1,r_{1}}^{†}M_{1,r_{1}}M_{2,r_{2}}]},$$
for combined measurement.

We describe a continuous measurement as a sequence of *n* measurements of duration δt, each one described by the Krauss operators in Equation (Equation 6). Each sequence of measurement produces a trajectory for the state of the coupled-qubit system. In order to calculate the average and variance of the heat extraction, we shall consider *N* different trajectories. Along each trajectory, we computed the exchanged heat for that particular sequence of measurement outcomes, taking into account only the stochasticity induced by the quantum measurements, and not by the stochastic nature of heat exchange with the baths [59,92].

In Figure 5, we compare the extracted heat distribution in the one-qubit measurement case (Panel (a)) and in the combined measurement case (Panel (b)) for *N* = 20,000 simulations of the heat extraction processes and for $F_{1}=-1$ in the left panel and F=(−1,−1) in the right panel. Each simulation was obtained by performing feedback after n=20 sequential measurements each of duration δt. Interestingly, in the case of a one-qubit continuous measurement, we found that the engine is more likely to extract zero heat, and the probability of extracting heat Q>0 decreases monotonically with *Q*. However, in the combined measurement case, a finite amount of heat (whose magnitude depends on the value of $\Delta_{z}$) is extracted more often than zero heat. The distribution in green is for the case when the two qubits are decoupled, whereas the blue distribution gives the finite coupling case $\left(\Delta_{z}=-0.1\;k_{B}T\right)$. We observed that the coupled system produces a larger average heat extraction (blue dashed line) compared to the decoupled system (green dashed line). However, the greater average heat extraction is accompanied by larger fluctuations, as observed from the broader width of the probability distribution for $\Delta_{z}=-0.1\;k_{B}T$. This can also be observed in Figure 6, where we plot the average heat extraction 〈Q〉 (upper panels) and the fluctuation quantified by the standard deviation $\sigma_{Q}$ (lower panels), as a function of $\Delta_{z}$. The panels on the left-hand side are for the one-qubit measurement case, whereas the right-hand side corresponds to the two qubits combined measurement case. We observed that the standard deviation reaches a minimum when the average extracted heat goes to zero. In addition, fluctuations are present both when the system is cooling and heating the environment. The maximum fluctuation is observed when the average heat extraction takes the maximum value. From the inset, we observe that the ratio between the average heat extracted and its standard deviation shows a maximum as a function of $\Delta_{z}$ in the 〈Q〉>0 regime. Comparing the one-qubit measurement and two qubits combined measurement cases, we observed that, although combined measurement gives better average heat extraction, it is also associated with larger fluctuations.

As we did for the discrete measurement case (see Figure 4), we now assess the impact of the combined quantum measurements. In Figure 7, we compare the “individual” and “combined” cases. The average heat extraction in the individual case is denoted with red circles, whereas the combined case is given by black crosses. The errors bars denote the standard deviation in the respective cases. As we observed in the discrete measurement case, there are system parameters (as the ones chosen in Figure 7) where the combined case outperforms the individual case. However, the larger heat extraction is also accompanied by larger fluctuations (compare the range of red and black error bars). This highlights once again the benefits of collective measurements for the average power of quantum thermal machines at the expense of larger fluctuations. This trade-off between power and power fluctuations is reminiscent of the thermodynamic uncertainty relations that have been derived, in the absence [93,94,95,96,97,98,99,100] and presence [101] of measurements, for quantum thermal machines.

## 4. Measurement-Assisted Refrigerator

In this section, we operate the continuously monitored coupled-qubit system as a measurement-assisted refrigerator. The two baths were considered non-detachable and will be kept at different temperatures to realize a refrigerator. More specifically, the refrigerator is powered by “swap operations” [102], which can be interpreted as work provided by a time-dependent driving that implements the unitary swap operation and by invasive quantum measurements in the absence of feedback. The state ρ of the coupled-qubits weakly coupled to the heat baths, under the influence of continuous measurements, is described by
(10)$$\frac{d\rho }{dt}=-i\left[H_{Q},\rho \right]+L_{B}\rho +L_{M}\rho ,$$
where […,…] represents the commutator. The first term in the right-hand side represents the unitary evolution of the system, whereas $L_{B}$ is a linear superoperator describing the dissipative dynamics induced by the coupling to the baths. To ensure the thermodynamic consistency of our results, the dissipative term $L_{B}\rho$ is derived using the global master equation [103], which satisfies the local detailed balance (see Appendix A for details). This guarantees that, for $T_{1}=T_{2}=T$ and in the absence of measurements and feedback, the state will evolve into thermal Gibbs state $\rho_{T}$. The third term on the right-hand side of Equation (Equation 10) is the quantum measurement contribution. It can be expressed as [47,48,86]
(11)$$L_{M}\rho =\Gamma_{M}D\left[X\right]\rho +\sqrt{\Gamma_{M}}H\left[X\right]\rho \frac{dW}{dt},$$
where $D\left[X\right]\rho =\left[X\rho X-\frac{1}{2}\left[XX\rho +\rho XX\right]\right]$ gives the dissipative contribution of the quantum measurement and $H\left[X\right]\rho =\left[X\rho +\rho X-2\langle X\rangle \rho \right]$ is the stochastic contribution. $\Gamma_{M}$ determines the strength of the measurement, and *X* is the system observable being measured. Only the first term survives upon averaging over the ensemble of measurement records. dW is a stochastic quantity, which results from the random nature of the measurements. The distribution for dW is Gaussian with zero mean and variance dt.

Let us denote the product of the eigenstates of $\sigma_{z}^{\left(i\right)}$ as {|0〉,|1〉,|2〉,|d〉}, where |0〉 represents the state where both qubits are in the ground state, |1〉 when only Qubit 1 is excited, |2〉 when only Qubit 2 is excited, and |d〉 when both qubits are excited. Motivated by the Hamiltonian of tunnel-coupled single-level quantum dot systems, where the doubly excited state may be energetically prohibited due to strong Coulomb interactions between the two quantum dots, we chose $E_{1}=\left(\epsilon_{1}-\tilde{\Delta }\right)/2$, $E_{2}=\left(\epsilon_{2}-\tilde{\Delta }\right)/2$, $\Delta_{z}=\tilde{\Delta }/2$, $\Delta_{x}=\Delta_{y}=\Delta /2$, and we considered the limit of large interaction $\tilde{\Delta }/\left(k_{B}T\right)$. We can thus neglect the |d〉 state, and Equation (Equation 1) for the coupled-qubit Hamiltonian reduces to
(12)$$H_{Q}=E_{1}\left|1\rangle \langle 1\right|+E_{2}\left|2\rangle \langle 2\right|+\Delta (|1\rangle \langle 2|+|2\rangle \langle 1|).$$The diagonalization of $H_{Q}$ leads to the basis $\left\{|0\rangle ,|+\rangle ,|-\rangle \right\}$, where the energy of the state |0〉 is zero, and the energy of the states |±〉 is
(13)$$E_{\pm }=\frac{E_{1}+E_{2}}{2}\pm \frac{1}{2}\sqrt{{\left(E_{1}-E_{2}\right)}^{2}+4\Delta^{2}}.$$The master equation in Equation (Equation 10) prescribes the evolution for the density matrix, which we express in the $\left\{|0\rangle ,|+\rangle ,|-\rangle \right\}$ basis in terms of transition rates and measurement parameters (see [64] for details). Here, we measured the state of $Q_{2}$ using $D_{2}$ by measuring the operator $\Pi_{X}=\left|2\rangle \langle 2\right|$.

In addition, after every measurement step of duration δt, we applied a unitary rotation given by
(14)$$U_{\mathrm{rot}}=\left[\begin{array}{cccc} 1 & 0 & 0\\ 0 & \cos\Theta & \sin\Theta & \\ 0 & \sin\Theta & -\cos\Theta \end{array}\right].$$For Θ=π/2, the unitary rotation $U_{\mathrm{rot}}$ becomes an effective swap gate $U_{\mathrm{SWAP}}$ between the |+〉 and |−〉 states.

Although we kept both diagonal and off-diagonal terms in our density matrix, we observed that, in the weak coupling and weak measurement limit, the contribution from the off-diagonal terms is very small compared to the contribution from the diagonal terms for $\Delta \ll E_{1},E_{2}$.

In Figure 8, we study the heat current $J_{2}$ flowing out of the bath at temperature $T_{2}\leq T_{1}$ when the coupled-qubit system is subject to continuous measurement and the swap operation after each measurement. The average heat flow out of the colder bath is given by the dashed black curve. The blue and red curves are obtained when only individual trajectories are considered and takes into account the stochastic nature of the measurement. We observed that, when $E_{1}>E_{2}$, the swap operation leads to a considerable cooling effect. However, when $E_{2}>E_{1}$, the swap operation leads to the heating effect.

In Figure 9a, we study $J_{2}$ as a function of rotation angle Θ for $E_{1}>E_{2}$. We observed that cooling is obtained even when there is no input work (Θ=0) [64] and maximum cooling is obtained for swap operation (Θ=π/2). The former is possible thanks to the invasive nature of quantum measurements, which changes the energetics of the quantum system, leading to a cooling effect upon an appropriate choice of the measurement [49,50]. The black dashed curve is obtained when there is input work without measurement, and the red curve is obtained when one considers both measurement and input work. We observed that the impact of continuous measurements on the heat current can be positive or negative depending on the rotation angle Θ. However, changing the value of parameter Δ, we observe in Figure 9b that a parameter regime exists where the combined effect of the invasive measurement and external work gives a better cooling effect for all values of Θ.

## 5. Conclusions

We studied coupled-qubit-based quantum thermal machines powered by quantum measurement and feedback. In the case of Maxwell’s demon, we studied various ways of implementing quantum measurement. We investigated both discrete and continuous measurement, as well as one-qubit measurement and two-qubit combined measurement. In the case of one-qubit measurement, and for a suitable choice of feedback, we observed that the heat extraction from the thermal bath increases monotonously as a function of $\sigma_{z}$-$\sigma_{z}$ coupling strength ($\Delta_{z}$) between the two qubits for a range of values of $\Delta_{z}$. We then compared the heat extracted from a single setup subject to combined measurements of both qubits, with the heat extracted from two setups operated in parallel, where only individual qubits are measured. Thanks to a collective effect, we found that the former can outperform the latter. In the case of continuous measurement, we studied the distribution of heat extraction for both one- and two-qubit measurement. Similar to the case of discrete measurement, in a certain parameter regime, we observed better average heat extraction with the combined measurement of two qubits compared to the individual measurement of each qubit in two parallel setups. However, better average heat extraction was always associated with higher fluctuations.

In the second part of the paper, we studied the measurement-assisted refrigeration in the coupled-qubit system attached to two thermal baths at different temperatures. We showed that although measurement and swap operations alone can power refrigeration, a combination of the two can yield higher refrigeration.

## Figures and Tables

**Figure 1 entropy-25-00204-f001:**
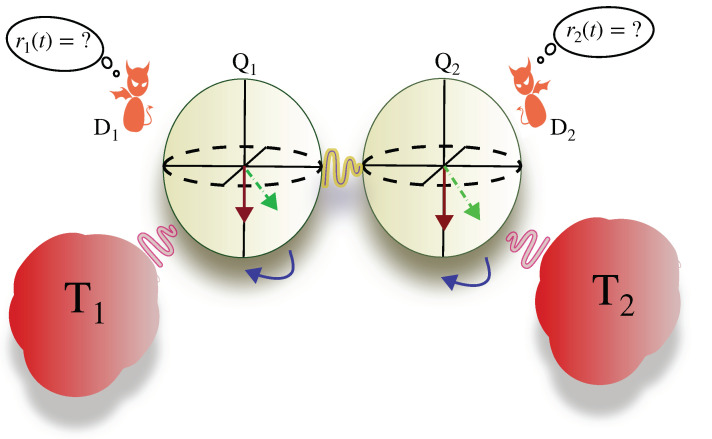
Coupled-qubit-based quantum feedback thermal machine. Qubit $Q_{i}$ is attached to a thermal bath with temperature $T_{i}$ and is being monitored by the measurement apparatus $D_{i}$ for i=1,2. In the case of Maxwell’s demon, $T_{1}=T_{2}=T$, whereas in the case of the measurement-assisted refrigerator, the two baths have different temperatures. Similarly, the two demons can undergo measurements with varying strengths.

**Figure 2 entropy-25-00204-f002:**
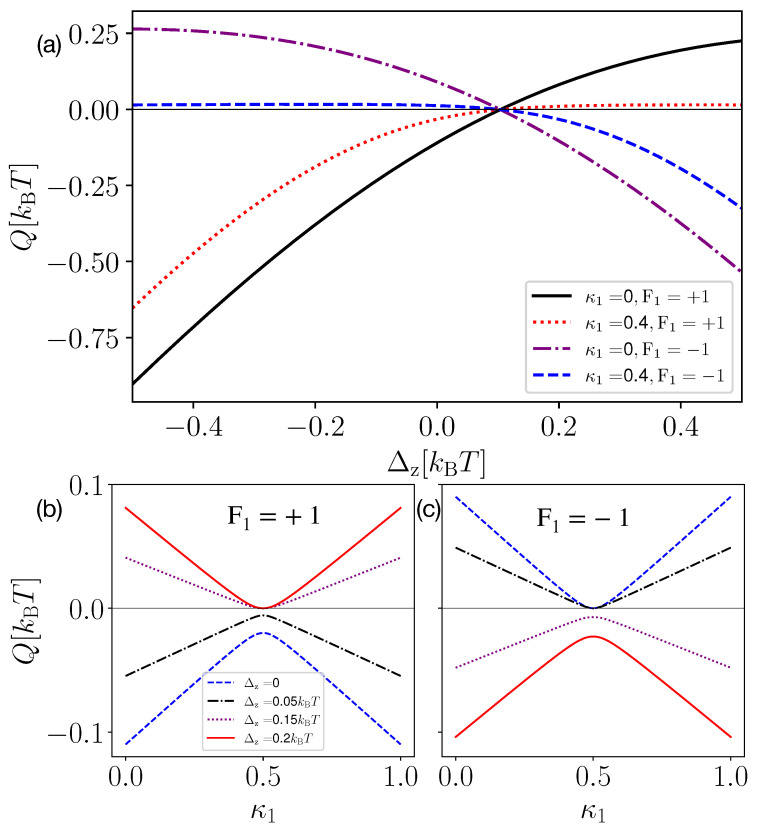
Heat extracted (Q) as a function of qubit–qubit coupling strength (Panel (**a**)) and measurement strength $\kappa_{1}$ (Panel (**b**) for $F_{1}=+1$ and Panel (**c**) for $F_{1}=-1$). In Panel (**a**), the black and red curves give the heat extraction for two values of the measurement strength $\kappa_{1}$ and for feedback $F_{1}=+1$ (rotation to positive *z*-axis). Similarly, the purple and blue curves give the heat extraction for feedback $F_{1}=-1$ (rotation to negative *z*-axis). In Panels (**b**,**c**), we plot the heat extraction as a function of $\kappa_{1}$ for $F_{1}=+1$ and $F_{1}=-1$, respectively, taking different coupling strengths between the qubits. We take $\epsilon_{1}=0.1\;k_{B}T$, $\epsilon_{2}=2\;k_{B}T$.

**Figure 3 entropy-25-00204-f003:**
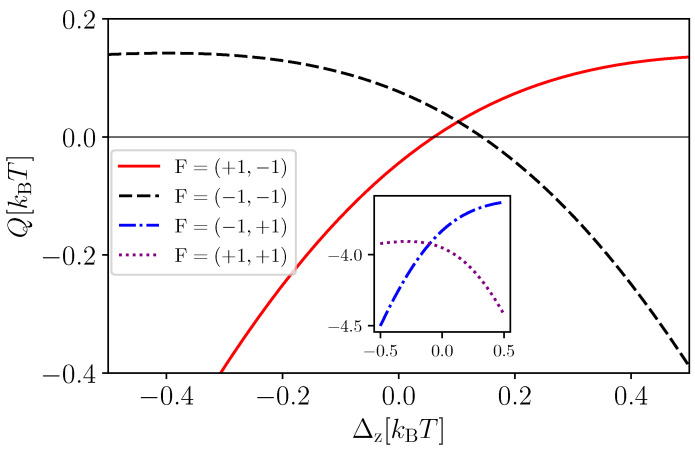
Heat extracted *Q* as a function of $\Delta_{z}$ for $\kappa_{1}=\kappa_{2}=0.2$. The feedback is represented as $F=(F_{1},\;F_{2})$, where $F_{i}$ is the feedback applied to the qubit $Q_{i}$. Finite heat extraction is obtained only when $F_{2}=-1$ (see the dashed black and solid red curves obtained with feedback F=(+1,−1) and F=(−1,−1), respectively). The dotted purple and dashed blue curves obtained with feedback F=(+1,+1) and F=(−1,+1) lead to the heating of the baths (see the inset). We take the same parameters as Figure 2.

**Figure 4 entropy-25-00204-f004:**
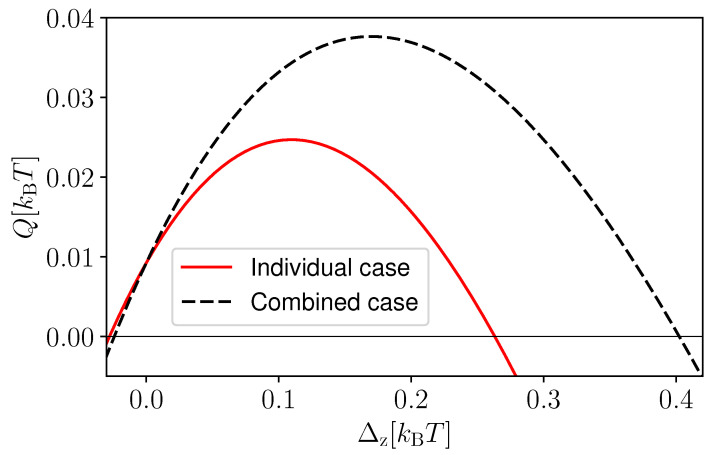
Heat extracted (*Q*) as a function of $\Delta_{z}$ for individual measurement (solid red curve) and combined measurement of two qubits (dashed black curve). As feedback, we applied F=(+1,−1) in both cases. We take $\epsilon_{1}=0.1\;k_{B}T$, $\epsilon_{2}=0.5\;k_{B}T$, $\kappa_{1}=\kappa_{2}=0.3$.

**Figure 5 entropy-25-00204-f005:**
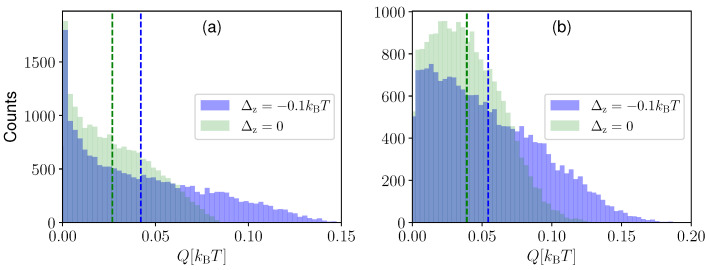
Count distribution of the heat extraction for one-qubit continuous measurement (Panel (**a**)) and the two-qubit combined continuous measurement (Panel (**b**)) for δt/τ=0.01. The dashed lines indicate the averages of the distributions. The simulation is performed for n=20 sequential measurements with feedback application only at the end. The distributions are for N= 20,000 simulations. As feedback, we applied $F_{1}=-1$ in the left panel and F=(−1,−1) in the right panel. We take the same parameters as Figure 2 for $\epsilon_{1},\epsilon_{2},k_{B}T$.

**Figure 6 entropy-25-00204-f006:**
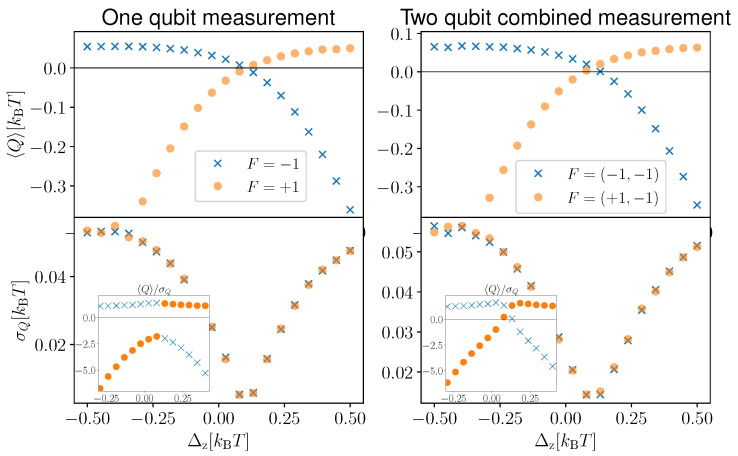
Average and standard deviation of the heat extraction for one-qubit continuous measurements (**left** panels) and two-qubit combined continuous measurements (**right** panel) for δt/τ=0.01. The simulation is performed for n=20 sequential continuous measurements with feedback application only at the end. The distributions are for N= 20,000 simulations. As feedback, we applied $F_{1}=-1$ in the left panel and F=(−1,−1) in the right panel. In the inset, we show the variation of the signal- (average heat extracted) to-noise (standard deviation of the extracted heat) ratio as a function of $\Delta_{z}$. We take the same parameters as Figure 2 for $\epsilon_{1},\epsilon_{2},k_{B}T$.

**Figure 7 entropy-25-00204-f007:**
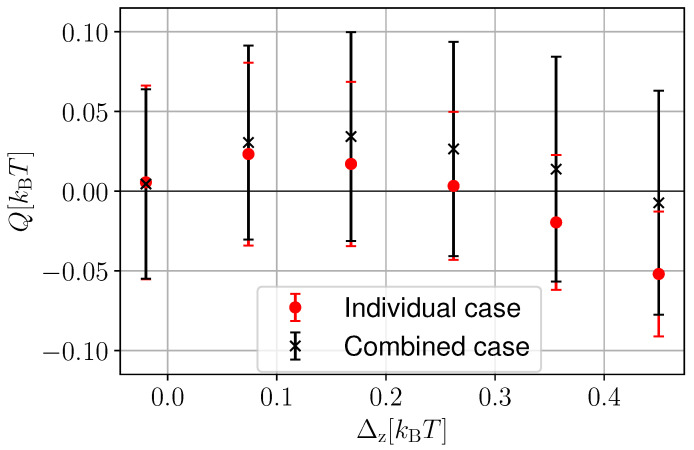
Heat extracted *Q* as a function of $\Delta_{z}$ in the individual (solid red curve) and the combined (dashed black curve) cases. The circles and crosses represent the average heat extracted 〈Q〉, whereas the error bars give the respective fluctuations $\sigma_{Q}$. We consider the number of measurements n=20 and the number of trajectories N=5000. As feedback, we applied F=(+1,−1) in both cases. We take $\epsilon_{1}=0.1\;k_{B}T$, $\epsilon_{2}=0.5\;k_{B}T$, δt/τ=0.01.

**Figure 8 entropy-25-00204-f008:**
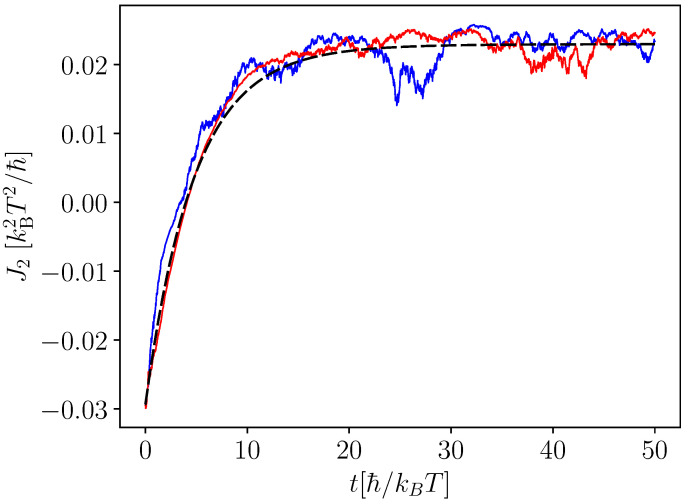
Refrigeration obtained as a result of measurement and a swap operation in the coupled-qubit system attached to two baths with different temperatures. The black dashed line gives the average heat current, whereas the red and blue curves are the heat current obtained for a single trajectory of measurement. We take $E_{1}=5\;k_{B}T$, $E_{2}=2\;k_{B}T$, $\Gamma_{1}=\Gamma_{2}=0.05$, $\Delta =0.2\;k_{B}T$, $\Gamma_{M}=0.02\;k_{B}T$, $T_{1}=1.1\;T$, $T_{2}=T$, and $\delta t=0.01\;\hbar /k_{B}T$. $\Gamma_{i}$ parameterizes the coupling strength between the coupled-qubit system and the bath i=1,2 (see Appendix A).

**Figure 9 entropy-25-00204-f009:**
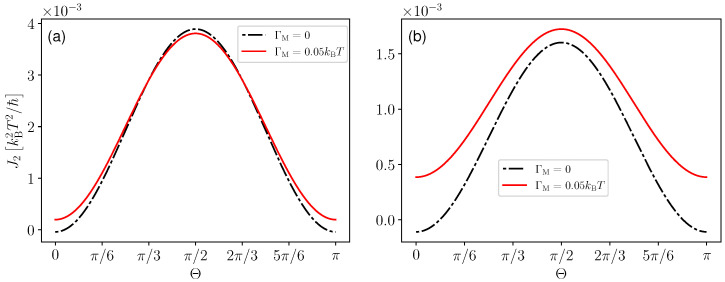
Heat current flowing out of the cold bath as a function of the rotation angle Θ for $\Delta =0.5\;k_{B}T$ (Panel (**a**)) and $\Delta =k_{B}T$ (Panel (**b**)). We take $T_{1}=1.1T$, $T_{2}=T$, $E_{1}=5k_{B}T$, $E_{2}=2k_{B}T$, $\Gamma_{1}=\Gamma_{2}=0.01$.

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
