# Peer review of "Measurement-Based Quantum Thermal Machines with Feedback Control"

_entropy, 2023, doi:10.3390/e25020204_

Round 1

Reviewer 1 Report

The authors analyze the impact of discrete and continuous measurements on the performance of quantum heat engines based on two qubits. They demonstrate that in feedback heat engine the performance (for both discrete and continuous measurements) is improved by performing join measurements on both qubits instead of separate ones; for the continuous case they also show that joint measurements increases not only the cooling power, but also heat fluctuations. Finally, they demonstrate that measurements can boost the power of swap-based heat engines.

The comparative analysis of joint a separate measurements is a novel and interesting result. Therefore, I recommend the publication after the following comments are addressed:

- The authors state that “The results obtained in this paper can be straightforwardly extended to the case of coupled quantum dots attached to fermionic baths”. However, this is not possible for feedback protocol analyzed in Sec. 3, since there is no analog of coherent qubit rotation around s_y axis in the fermionic case (an analogous operation is forbidden by the parity superselection rules which prohibits coherent superpositions of states with an even and odd occupancy). The authors should clarify their statement or omit it completely.

- The Hamiltonian (12) looks like it include s_x^1 s_y^1 + s_x^2 s_y^2 interaction rather than s_x^1 s_x^2 [as defined in Eq. (1)].

- In Fig. 6 it would be informative to plot also the ratio of the standard deviation to the average heat.

Reviewer 2 Report

Comments can be found in the attachment.
